**Subject Category:**
Biology (whole organism)

ecology/evolution

phage–bacteria infection network, Vibrio, filamentous phages, salinity changes

**Author for correspondence:**
Carolin C. Wendling
e-mail: carolin.wendling@env.ethz.ch

# Filamentous phages reduce bacterial growth in low salinities

Henry Goehlich[1], Olivia Roth[1] and Carolin C. Wendling[1,2]

[1]Marine Evolutionary Ecology, GEOMAR Helmholtz Centre for Ocean Research Kiel, Düsternbrooker Weg 20, 24105 Kiel, Germany
[2]ETH Zürich, Institute of Integrative Biology, Universitätstrasse 16, CHN D 33, 8092 Zürich, Switzerland

 HG, 0000-0002-6032-2760; OR, 0000-0002-7349-7797;
CCW, 0000-0003-4532-7528

Being non-lytic, filamentous phages can replicate at high frequencies and often carry virulence factors, which are important in the evolution and emergence of novel pathogens. However, their net effect on bacterial fitness remains unknown. To understand the ecology and evolution between filamentous phages and their hosts, it is important to assess (i) fitness effects of filamentous phages on their hosts and (ii) how these effects depend on the environment. To determine how the net effect on bacterial fitness by filamentous phages changes across environments, we constructed phage–bacteria infection networks at ambient 15 practical salinity units (PSU) and stressful salinities (11 and 7 PSU) using the marine bacterium, *Vibrio alginolyticus* and its derived filamentous phages as model system. We observed no significant difference in network structure at 15 and 11 PSU. However, at 7 PSU phages significantly reduced bacterial growth changing network structure. This pattern was mainly driven by a significant increase in bacterial susceptibility. Our findings suggest that filamentous phages decrease bacterial growth, an indirect measure of fitness in stressful environmental conditions, which might impact bacterial communities, alter horizontal gene transfer events and possibly favour the emergence of novel pathogens in environmental *Vibrios*.

## 1. Introduction

Bacteriophages are highly abundant in the oceans with profound implications on bacterial ecology and evolution [1]. Phages increase bacterial population diversity [2,3], affect bacterial competition [4], mediate horizontal gene transfer [5,6] and play a major role in the marine nutrient cycle [7]. Identifying who infects whom, and how phages can change the fitness of their host, is crucial to predict phage-mediated impacts on microbial communities.

Phage–bacteria infection networks (PBINs) are ideal to identify phage host range, bacterial resistance and their co-evolutionary dynamics [8]. Previous studies focused mainly on infection networks between bacteria and lytic phages [9–11]. By contrast, infection networks between bacteria and temperate or filamentous phages, are just beginning to be explored [12,13].

Being non-lytic, filamentous phages constitute an antithesis to the classical view of head-and-tail killer phages [14]. Filamentous phages do not form classical host–parasite relationships with their hosts, but have evolved to keep the balance between their own reproduction and host survival [15]. Thus, the net effect on bacterial fitness imposed by filamentous phages varies, depending on ecological and evolutionary factors. For instance, high-efficiency-replicating phages like Ff of *Escherichia coli* that do not bring any benefit to their host, seem to be more parasitic than low-efficiency-replicating phages, which contribute to bacterial virulence through accessory virulence genes, for instance, CTX-phage of *Vibrio cholerae* [15].

Bacteria of the genus *Vibrio* harbour a diverse repertoire of filamentous phages, which play an important role in the evolution and emergence of novel pathogens [16,17]. The host range of filamentous vibriophages is correlated to the phylogenetic distance of the phage's host bacterium and all other bacteria that they are able to infect [12]. But, even if isolated from closely related host bacteria the host range of filamentous phages can differ significantly among individual phages [13].

Crucially, studies on lytic phages revealed that phage host range is rarely a binary trait and that it is important to take the abiotic environment into consideration to infer the underlying ecological and evolutionary dynamics [18]. Abiotic factors can influence the phage infection and replication process: for instance, a reduction in salinity can alter the adsorption rates of various phages onto their bacterial host, which changes infection outcomes between bacteria and phages [19]. Thus, in the light of the phage-driven emergence of novel *Vibrio* pathogens, we set out to investigate whether and in which direction abiotic factors, in particular salinity, affect the structure of filamentous PBINs.

Marine bacteria of the genus *Vibrio* are present along the coast of the Baltic Sea [20,21] and their abundance, genetic composition and virulence are related to environmental factors, including temperature and salinity [20,22,23]. Characteristic for this habitat is a salinity gradient ranging from almost freshwater in the North and East to nearly fully marine conditions at the transition to the North Sea [24]. By the end of the century, the average salinity in the Baltic Sea is predicted to drop by approximately 4 practical salinity units (PSU) as a result of global climate change [25]. Thus, the Baltic Sea represents a natural time machine for the future coastal ocean [26] making it an ideal ecosystem to study the effect of environmental change on species interactions.

In this study, we investigated how reduced salinities influence the interaction between bacteria and filamentous phages using an established model system, i.e. the marine bacterium *Vibrio alginolyticus* and filamentous phages thereof. We measured the reduction in bacterial growth (RBG) imposed by filamentous phages at ambient salinity and at two different reduced salinities and compared networks from the different salinities to identify the drivers causing the different network structures.

# 2. Methods

## 2.1. Bacteria isolation

All *V. alginolyticus* strains were isolated from nine healthy broad-nosed pipefish (*Syngnathus typhle*) caught in the Kiel Fjord (ambient salinity of 15 PSU) [21]. For the present study, we selected a subset of 32 out of 75 previously characterized bacterial strains comprising three different phage-susceptibility categories [13]: two highly susceptible, 28 intermediate susceptible and three resistant strains, as well as their derived filamentous phages, to investigate whether the PBIN changes with different salinities. We chose three different salinity levels: current ambient salinity conditions in the Kiel Fjord region (15 PSU), predicted average salinity conditions (11 PSU) and potential temporary salinity conditions at the end of the century (7 PSU) [25,27].

## 2.2. Phage–bacteria infection assay

### 2.2.1. Phage extraction

Filamentous phages were obtained from the supernatant of bacterial cultures during the exponential phase. To control for variations in phage production across salinities, we decided to extract all phages at 15 PSU and only perform the infection assays at the three different salinities. All *Vibrio* strains were

revived from cryo cultures in liquid Medium101 (Medium101: 0.5% (w/v) peptone, 0.3% (w/v) meat extract, 1.5% (w/v) NaCl in MilliQ water) overnight at 25°C and constant shaking at 180 r.p.m. Subsequently, overnight cultures were diluted 1 : 100 and grown for 7 h in 15 ml Falcon tubes with 4 ml Medium101 at 25°C and 180 r.p.m. Afterwards, bacterial cultures were centrifuged at 6000*g* for 10 min resulting in a bacterial pellet and suspended phages in the supernatant. Phage-containing supernatant was filtered (pore size: 0.20 µm) to remove all bacteria and 10-fold diluted in TM buffer (modified from Sen & Ghosh [28]: 50% (v/v) 20 mM $MgCl_2$, 50% (v/v) 50 mM Tris–HCl, pH 7.5).

The presence of active filamentous phages in these supernatants has been confirmed in an earlier study [12]. To re-confirm the presence of phages in the current study, we performed spot assays on a highly susceptible host strain K01M1, as described in [13]. In addition, a subset of nine strains has been fully sequenced, which revealed the presence of one to two different filamentous phages per strain [13]. Based on these observations, we are confident that the supernatants in the present study contain filamentous phages.

### 2.2.2. Reduction in bacterial growth assay

We measured the RBG imposed by the phage in liquid culture [29] for more than 9000 possible phage–bacteria interactions (32 phages × 32 bacteria × 3 salinities × 3 replicates) with media and growth time adjusted to the Baltic *Vibrio*-phage system [12]. To do so, overnight cultures of all bacterial strains (grown at 15 PSU) were diluted 1 : 10 in their respective Medium101 at 7, 11 or 15 PSU and incubated with constant shaking at 180 r.p.m. and 25°C for 1.5 h (15 and 11 PSU) or 2 h (7 PSU), respectively, to achieve similar starting optical densities (OD600) ranging between 0.104 and 0.125. We added 15 µl of bacteria in the log phase (concentration: $5 \times 10^6$ cells $ml^{-1}$) to transparent, flat-bottom 96-well microtiter (Nunclon™) plates containing 120 µl Medium101 with the respective salinity and either 15 µl TM buffer containing phage lysate (with a $10^{-1}$ dilution of the original phage-containing supernatant) or 15 µl TM buffer only as control. We measured bacterial optical density at 600 nm (OD600) using an automated plate reader at the beginning ($t = 0$ h) and after 18 h ($t = 18$ h) of static incubation at 25°C. The RBG imposed by phage *i* on strain *j*, was calculated according to the following formula:

$$RBG_{ij} = \frac{OD600\ (t = 18\,h) - OD600\ (t = 0\,h)\ [ij]}{OD600\ (t = 18\,h) - OD600\ (t = 0\,h)\ [\text{Control } j]}.$$

The calculation for the threshold of infection was adapted from Poullain *et al.* [29] and revealed a bimodal histogram of all RBG values with a local minimum at RBG = 0.82 for our *Vibrio* system [2]. We thus concluded that RBG values lower than 0.82 indicate a significant RBG imposed by the phage and thus a successful infection, which has a negative impact on bacterial fitness. RBG values higher than 0.82 do not necessarily indicate the absence of an infection, but they indicate that the phage does not have a negative impact on bacterial fitness.

Due to handling errors during the RBG assay, we had to remove three phages (ΦK09K2, ΦK09K3 and ΦK10K4) from the analysis.

## 2.3. Bacterial growth rates and phage production

### 2.3.1. Bacterial growth rates

We measured bacterial growth at each salinity to assess potential correlations between growth rates and phage infections by means of 24 h growth curves. Growth curves were obtained for each strain at each salinity using 96-well plates. In brief, overnight cultures of each strain were diluted 1 : 100 with Medium101 containing the respective salinity (7, 11 or 15 PSU), and 200 µl of these dilutions were transferred to a 96-well plate. Optical density was measured in triplicates at 600 nm every 15 min over 24 h using an automated plate reader.

### 2.3.2. Phage production

To assess whether salinity influences the phage production of resident phages, we measured the amount of plaque-forming units relative to colony-forming units (PFU/CFU) for each strain. CFU was measured by plating 100 µl of $10^{-6}$ and $10^{-7}$ diluted overnight cultures onto thiosulfate-citrate-bile salts-sucrose (TCBS) agar plates, and the number of colonies was counted after overnight incubation at 25°C. PFU was measured by small-scale polyethylene glycol (PEG) precipitation of the same overnight cultures used for quantifying CFU, because quantification via standard spot assays was not possible as we mostly observed opaque zones of reduced growth. Thus, we used spectrometry to quantify phage

prevalence, which uses the constant relationship between the length of viral DNA and the amount of the major coat protein VIII of filamentous phages, which, together, are the main contributors of the absorption spectrum in the UV range (http://www.abdesignlabs.com/technical-resources/bacteriophage-spectrophotometry). The amount of phage particles per millilitre can be calculated according to the following formula:

$$\text{phages ml}^{-1} = \frac{(\text{OD269} - \text{OD320}) * 6e16}{\text{bp}},$$

where OD269 and OD320 stand for optical density at 269 and 320 nm and bp stands for a number of base pairs per phage. Phage length used for calculation was 7079 bp. After centrifuging 1500 µl of the phage-containing overnight culture at 13 000$g$ for 2 min, 1200 µl of the supernatant was mixed with 300 µl PEG/NaCl five times and incubated on ice for 30 min. Afterwards, phage particles were pelleted by two rounds of centrifugation at 13 000$g$ for 2 min, resuspended in 120 µl Tris-buffered saline (TBS) once and incubated on ice. After 1 h, the suspension was cleaned by centrifugation at 13 000$g$ for 1 min and absorbance was measured at 269 and 320 nm.

## 2.4. Statistical analysis

### 2.4.1. Network analysis

After confirming that all three technical replicates per salinity did not differ significantly (Mantel test; Monte Carlo test observation based on 9999 permutations: $p < 0.001$), we created a consensus matrix for each salinity (positive infection: infection occurs at least in two replicates, electronic supplementary material, S1). We performed network analyses for each consensus matrix based on the Falcon interactive mode [30] and the bipartite package [31] using nested measures based on the overlap and decreasing fill (NODF) and the SS null model [32]. Analysis of nestedness can be sensitive to the inclusion of empty rows and columns, we thus performed the analysis again after removing all empty rows and columns of each network.

The impact of salinity on network connectance and fill, the number of infecting phages and the number of susceptible bacteria was analysed using a linear model for each of these parameters with salinity as fixed effect.

### 2.4.2. Bacterial growth rates

We used the grofit package [33] to calculate growth rates ($\mu$) from 24 h growth curves. To determine differences in growth rates between different salinities and strains, we used a linear mixed-effect model (package: lme4) [34] with salinity and strain as fixed factors and the day of measurement as random effect.

### 2.4.3. Correlation of phage infections with bacterial growth rates

We used ggscatter (package: ggpubr) to test for a correlation between growth rate at 7 PSU relative to 15 PSU and the number of phages causing infections at 7 PSU in a single bacterial strain. We further tested for correlations between the number of phages causing infections in a single bacterial strain and the respective growth rates at 7, 11 and 15 PSU.

### 2.4.4. Production of resident phages

To determine whether resident phages of each strain produce different amounts of phages in the different salinities, we performed a paired $t$-test between PFU/CFU at 7 PSU and PFU/CFU at 15 PSU.

# 3. Results and discussion

## 3.1. Filamentous phages reduce bacterial growth at low salinities

Based on three-times-replicated PBIN (figure 1), we compared network structures of filamentous PBINs across three different salinities, relevant for the Kiel Fjord region: current average salinity with 15 PSU, current annual minimum salinity and predicted average salinity by the end the of the century (11 PSU) as well as predicted short-term minimum salinity by 2100 (7 PSU). One limitation of the present study is that we cannot infer cryptic infections, i.e. infections, which do not reduce bacterial growth. Thus, we

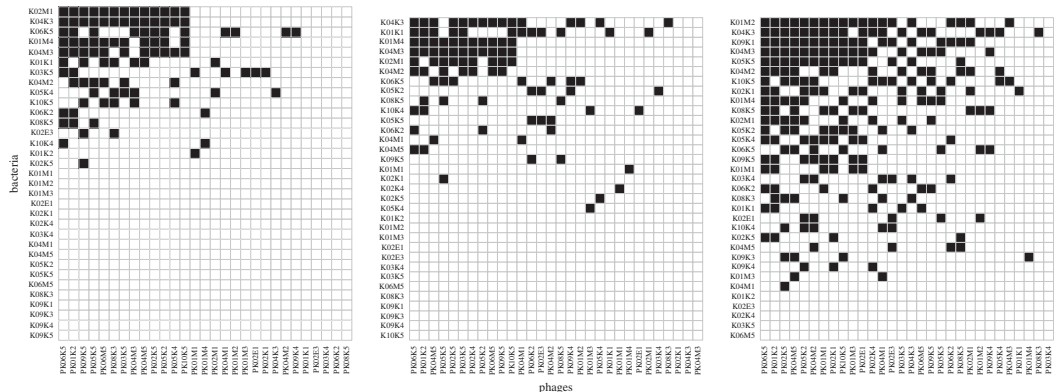

**Figure 1.** Phage–bacteria infection network (PBIN) of the consensus matrices. Rows: bacteria, columns: phages, black cells: successful infection. 15 PSU (left), 11 PSU (middle) and 7 PSU (right).

**Table 1.** Network statistics at 15, 11 and 7 PSU of the consensus matrices. Nestedness based on the overlap and decreasing fill (NODF: 100 = perfectly nested and 0 = zero nestedness), $nT$: normalized temperature, indicating the deviation from the null model (less than 1 observed network is less nested than expected). Connectance: ratio of the number of successful infections relative to all possible interactions between phages and bacteria. Fill: total amount of successful infections.

| salinity (PSU) | NODF | $nT$ | Z-score | p-value | connectance | susceptible bacteria | infecting phages | fill |
|---|---|---|---|---|---|---|---|---|
| 15 | 41.88 | 1.54 | 7.24 | 0.001 | 0.11 | 16 | 24 | 103 |
| 11 | 29.8 | 1.59 | 7.7 | 0.001 | 0.10 | 20 | 27 | 101 |
| 07 | 51.22 | 1.65 | 14.73 | 0.001 | 0.25 | 28 | 29 | 246 |

only consider instances, where infections are harmful for the bacterium, as in reducing its growth rate by more than 20%, as true infections.

All PBINs showed a significant nested structure based on the overlap and decreasing fill (NODF, table 1) with the highest nestedness at 7 PSU (51.22) followed by 15 PSU (41.9), and lowest nestedness at 11 PSU (29.8) suggesting that the nested structure remains consistent irrespective of the salinity level. We did not find a difference in nestedness measures after excluding empty rows and columns from the analysis, confirming that all three networks are truly nested, which was previously observed in filamentous PBINs containing closely related *Vibrio* strains [12,13].

Network parameters were similar for the 11 and 15 PSU networks but changed significantly at 7 PSU (figure 2, table 2). This indicates that unlike a gradual increase, a certain salinity threshold is required to cause this significant RBG in the presence of filamentous phages (table 2).

The number of phages able to at least infect one bacterium did not differ among all three salinities. Instead, the number of bacteria susceptible to at least one phage infection almost doubled from 15 to 7 PSU (figure 2). Thus, the observed RBG at 7 PSU seems to be mainly driven by increased bacterial susceptibility rather than phage infectivity.

The aim of this study was to determine how salinity changes the impact of filamentous phages on bacterial growth and thus ultimately bacterial fitness and the network structure of filamentous PBINs. In the future, the underlying molecular mechanisms, which could explain the observed decrease in bacterial growth at 7 PSU, should be identified. Thus, we can only speculate that various parameters, for instance, differences in phage adsorption rates (salinity can affect the adsorption to the bacterial cell wall in both directions [9,19,35–37]) can cause the observed change in network structure in the present study.

Reduced salinities represent a stressful condition for *V. alginolyticus* and result in a reduction in growth rate ($\mu$) by up to 50% relative to ambient conditions ($\chi_2^2 = 2641.2$, $p < 0.001$) and overall prolonged growth (electronic supplementary material, S2). This negative impact on bacterial growth parameters could have influenced the infection outcome at low salinities in several ways. It is, for instance, possible that the prolonged growth at 7 PSU might increase the time window for phage infection at low salinities. Future studies would have to compare the number of phage infections at

**Figure 2.** Comparison of selected network parameters at 15, 11 and 7 PSU. (*a*) network connectance, (*b*) network fill, (*c*) number of susceptible bacteria, (*d*) number of infecting phages. Significant differences are indicated by small letters.

**Table 2.** Linear model for selected network parameters: C: network connectance, F: network fill, P: number of infecting phages, B: number of susceptible bacteria. Shown are comparisons for 11 PSU and 7 PSU relative 15 PSU (intercept). ***$p < 0.001$.

|   |           | estimate | s.e. | *t*-value | *p* (>*t*) |
|---|-----------|----------|------|-----------|------------|
| C | intercept | 0.13     | 0.01 | 18.52     | $1.60 \times 10^{-6}$*** |
|   | 11 PSU    | −0.02    | 0.01 | −1.64     | 0.15 |
|   | 7 PSU     | 0.15     | 0.01 | 14.40     | $7.02 \times 10^{-6}$*** |
| F | intercept | 130.00   | 6.97 | 18.66     | $1.53 \times 10^{-6}$*** |
|   | 11 PSU    | −14.67   | 9.85 | −1.49     | 0.19 |
|   | 7 PSU     | 139.33   | 9.85 | 14.14     | $7.80 \times 10^{-6}$*** |
| P | intercept | 28.33    | 0.64 | 44.39     | $8.75 \times 10^{-9}$*** |
|   | 11 PSU    | −0.33    | 0.90 | −0.37     | 0.73 |
|   | 7 PSU     | 0.33     | 0.90 | 0.37      | 0.73 |
| B | intercept | 19.00    | 0.61 | 31.22     | $7.17 \times 10^{-8}$*** |
|   | 11 PSU    | 2.00     | 0.86 | 2.32      | 0.06 |
|   | 7 PSU     | 10.67    | 0.86 | 12.39     | $1.68 \times 10^{-5}$*** |

selected time points during 24 h at all three salinities. Furthermore, it is possible that resident filamentous phages produce at higher frequencies at 7 PSU, which is costly for the bacterial host and will thus result in reduced growth rates. However, we did not find a significant difference in phage production between 7 and 15 PSU salinities (paired *t*-test: $t_{32} = 1.16$, $p = 0.255$, electronic supplementary material, S3) and can thus exclude that reduced growth at 7 PSU is a direct consequence of costly phage production. Lastly, we tested whether the reduction in growth alone explains the increased number of phage infections at 7 PSU. To do so, we determined whether those bacterial strains that experienced a strong reduction in growth at 7 PSU relative to 15 PSU are driving the increased number of successful phage infections at 7 PSU. In contrast to this expectation, strain-specific reduction of growth rate at 7 PSU relative to 15

PSU was not an indicator for the observed increase in successful phage infections ($r = 0.065$, $p = 0.52$, electronic supplementary material, S4). Furthermore, there was no correlation between the number of phage infections per strain and strain-specific growth rate at the three salinity levels (electronic supplementary material, S5), which differed significantly between strains ($\chi^2_{32} = 477.8$, $p < 0.001$). This indicates that reduced bacterial growth alone at 7 PSU is not an indicator for the observed increase in successful infections.

## 3.2. Ecological and evolutionary implications

We show that filamentous phages significantly decrease bacterial growth at low salinities, a pattern that is predominantly driven by increased bacterial susceptibility rather than increased phage infectivity. According to the present data, we do not expect an increase in successful infections of *Vibrios* by filamentous phages in response to the predicted reduction of the average salinity concentration from 15 to 11 PSU in the Kiel Fjord by the end of the century [25]. However, due to saltwater inflow events from the North Sea, rainfall and river runoffs, salinity concentrations in the coastal area of the southern Baltic Sea usually fluctuate by five units above and below the average salinity level, which can result in rapid salinity drops down to 7 PSU and below [27]. To this end, these temporal low salinity events may result in an increase in filamentous phage infections, which can have various consequences at population and community levels. For instance, if these infecting phages carry virulence genes, which is common for filamentous vibriophages [16,17], new pathogenic *Vibrios* will possibly emerge in response to decreasing salinities. To study these long-term interactions between *Vibrios* and filamentous phages in different environmental conditions, future work should focus on how abiotic parameters influence the interaction of filamentous phages and *Vibrio* bacteria on evolutionary time scales. It might be possible, that bacteria will adapt to counter these negative effects resulting from increasing phage pressure at low salinities. Overall, our results confirm that species interactions are strongly shaped by the environment and that it is paramount to integrate abiotic factors when studying them.

Data accessibility. Data were deposited at PANGAEA. https://doi.pangaea.de/10.1594/PANGAEA.906121.
Authors' contributions. C.C.W., O.R. and H.G. designed the study. H.G. performed the experiments, analysed the data and wrote the first draft. C.C.W., H.G. and O.R. wrote the final version.
Competing interests. We declare we have no competing interests.
Acknowledgements. We thank Angela Stippkugel and Kim-Sara Wagner for their support in the laboratory. This project was funded by a DFG grant no. (WE 5822/1–1) within the priority programme SPP1819 given to C.C.W. and O.R. We thank all anonymous reviewers for valuable comments on the manuscript and the International Max Planck Research Schools for career and financial support of H.G.

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
