## [Reviewer comments · Royal Society Open Science]

Review History

RSOS-191669.R0 (Original submission)

Review form: Reviewer 1

Is the manuscript scientifically sound in its present form?

Yes

Are the interpretations and conclusions justified by the results?

Yes

Is the language acceptable?

Yes

Do you have any ethical concerns with this paper?

No

Have you any concerns about statistical analyses in this paper?

No

Recommendation?

Accept with minor revision (please list in comments)

Comments to the Author(s)

In this paper, the authors test phage host range under varying salinities, using conditions that represent possible variation during regular fluctuations as well as under climate change. Overall, this is a well-written paper with interesting results. I have a few minor questions/concerns listed below:

Minor concerns:

1. Depending on the journal space requirements, I would recommend moving more (or all) of the supplementary methods into the main text. Similarly, I would recommend restating more of the essential method details that are described in other papers (e.g. that phages were originally induced with Mitomycin C and diluted in TM buffer). This would be more convenient for the reader.
2. How reliable is the growth rate reduction assay at all salinities? Is it possible that phages are more harmful at low salinity but susceptibility is actually not higher? I.e. could you have cryptic infections that do not infect at high rates or do not infect with high virulence unless at low salinity? One possibility is that the stress from the low salinity leads to a higher rate of spontaneous phage induction, and that phages do infect at 15 PSU but do so without causing significant loss of host growth. If possible, I would suggest doing PCR assays on a random subsample of both negative and positive results to confirm the growth curve assay inferences.
3. Because the 7 PSU treatments grew slower, the authors varied the timing of the infection assay to try to do infections at the same relative part of the bacterial growth curves. Looking at the supplementary figure with bacterial growth curves, low salinity not only had slower growth but prolonged growth (and in some graphs what appears to be diauxic growth). Is it possible that these changes in timing, though, might explain some of the differences in bacterial susceptibility? That is to say, might low salinity increase the time window for phage infection?
4. Very minor semantics issue: In figure S6, a label refers to “number of phages causing lytic infections.” The use of “lytic” here seems inappropriate since the assay does not distinguish between reduced growth and increased mortality (and since the phages are filamentous).

Review form: Reviewer 2

Is the manuscript scientifically sound in its present form?

Yes

Are the interpretations and conclusions justified by the results?

Yes

Is the language acceptable?

Yes

Do you have any ethical concerns with this paper?

No

Have you any concerns about statistical analyses in this paper?

Yes

Recommendation?

Accept with minor revision (please list in comments)

Comments to the Author(s)

I enjoyed reading this paper. An understanding of how abiotic factors influence bacteria-phage networks is one of fundamental and applied interest. I have a couple of minor suggestions:

I am glad that the authors have pointed out that studying the interactions over a longer (evolutionary time-scales) is required. One could predict that bacteria will adapt to counter the negative effects on fitness at lower salinities. With this point in mind I think the authors should scale back or tone down the language relating to the potential consequences of this reduced salinity in the future with respect to the evolution and emergence of pathogens i.e. last sentence of the abstract and Line 298-299 e.g. "we can expect the emergence of new pathogenic Vibrios in response to decreasing salinities." The use of the word "can" expect sounds as though this outcome is almost guaranteed... Similarly, I think the final sentence of the abstract is overstated based on the actual findings presented.

Spell out PSU in the first instance

Line 181 check font

Decision letter (RSOS-191669.R0)

18-Oct-2019

Dear Dr Wendling

On behalf of the Editors, I am pleased to inform you that your Manuscript RSOS-191669 entitled "Filamentous phages reduce bacterial growth in low salinities" has been accepted for publication in Royal Society Open Science subject to minor revision in accordance with the referee suggestions. Please find the referees' comments at the end of this email.

The reviewers and handling editors have recommended publication, but also suggest some minor revisions to your manuscript. Therefore, I invite you to respond to the comments and revise your manuscript.

- Ethics statement

- Data accessibility

It is a condition of publication that all supporting data are made available either as supplementary information or preferably in a suitable permanent repository. The data accessibility section should state where the article's supporting data can be accessed. This section should also include details, where possible of where to access other relevant research materials such as statistical tools, protocols, software etc can be accessed. If the data has been deposited in an external repository this section should list the database, accession number and link to the DOI for all data from the article that has been made publicly available. Data sets that have been

deposited in an external repository and have a DOI should also be appropriately cited in the manuscript and included in the reference list.

If you wish to submit your supporting data or code to Dryad (<http://datadryad.org/>), or modify your current submission to dryad, please use the following link:
<http://datadryad.org/submit?journalID=RSOS&manu=RSOS-191669>

- **Competing interests**

- **Authors' contributions**

- **Acknowledgements**

- **Funding statement**

Because the schedule for publication is very tight, it is a condition of publication that you submit the revised version of your manuscript before 27-Oct-2019. Please note that the revision deadline will expire at 00.00am on this date. If you do not think you will be able to meet this date please let me know immediately.

When submitting your revised manuscript, you will be able to respond to the comments made by

the referees and upload a file "Response to Referees" in "Section 6 - File Upload". You can use this to document any changes you make to the original manuscript. In order to expedite the processing of the revised manuscript, please be as specific as possible in your response to the referees. We strongly recommend uploading two versions of your revised manuscript:

If your manuscript is newly submitted and subsequently accepted for publication, you will be asked to pay the article processing charge, unless you request a waiver and this is approved by Royal Society Publishing. You can find out more about the charges at <http://rsos.royalsocietypublishing.org/page/charges>. Should you have any queries, please contact opscience@royalsociety.org.

Kind regards,
Anita Kristiansen

Editorial Coordinator
Royal Society Open Science
openscience@royalsociety.org

on behalf of Dr Berat Haznedaroglu (Associate Editor) and Kevin Padian (Subject Editor)
openscience@royalsociety.org

Reviewer comments to Author:

Reviewer: 1

Comments to the Author(s)

In this paper, the authors test phage host range under varying salinities, using conditions that represent possible variation during regular fluctuations as well as under climate change. Overall, this is a well-written paper with interesting results. I have a few minor questions/concerns listed below:

Minor concerns:

1. Depending on the journal space requirements, I would recommend moving more (or all) of the supplementary methods into the main text. Similarly, I would recommend restating more of the essential method details that are described in other papers (e.g. that phages were originally induced with Mitomycin C and diluted in TM buffer). This would be more convenient for the reader.
2. How reliable is the growth rate reduction assay at all salinities? Is it possible that phages are more harmful at low salinity but susceptibility is actually not higher? I.e. could you have cryptic infections that do not infect at high rates or do not infect with high virulence unless at low salinity? One possibility is that the stress from the low salinity leads to a higher rate of spontaneous phage induction, and that phages do infect at 15 PSU but do so without causing significant loss of host growth. If possible, I would suggest doing PCR assays on a random subsample of both negative and positive results to confirm the growth curve assay inferences.
3. Because the 7 PSU treatments grew slower, the authors varied the timing of the infection assay to try to do infections at the same relative part of the bacterial growth curves. Looking at the supplementary figure with bacterial growth curves, low salinity not only had slower growth but prolonged growth (and in some graphs what appears to be diauxic growth). Is it possible that these changes in timing, though, might explain some of the differences in bacterial susceptibility? That is to say, might low salinity increase the time window for phage infection?
4. Very minor semantics issue: In figure S6, a label refers to "number of phages causing lytic infections." The use of "lytic" here seems inappropriate since the assay does not distinguish between reduced growth and increased mortality (and since the phages are filamentous).

Reviewer: 2

Comments to the Author(s)

I enjoyed reading this paper. An understanding of how abiotic factors influence bacteria-phage networks is one of fundamental and applied interest. I have a couple of minor suggestions:

I am glad that the authors have pointed out that studying the interactions over a longer (evolutionary time-scales) is required. One could predict that bacteria will adapt to counter the negative effects on fitness at lower salinities. With this point in mind I think the authors should

scale back or tone down the language relating to the potential consequences of this reduced salinity in the future with respect to the evolution and emergence of pathogens i.e. last sentence of the abstract and Line 298-299 e.g. "we can expect the emergence of new pathogenic Vibrios in response to decreasing salinities." The use of the word "can" expect sounds as though this outcome is almost guaranteed... Similarly, I think the final sentence of the abstract is overstated based on the actual findings presented.

Spell out PSU in the first instance
Line 181 check font

Author's Response to Decision Letter for (RSOS-191669.R0)

See Appendix A.

Decision letter (RSOS-191669.R1)

13-Nov-2019

Dear Dr Wendling,

It is a pleasure to accept your manuscript entitled "Filamentous phages reduce bacterial growth in low salinities" in its current form for publication in Royal Society Open Science. The comments of the reviewer(s) who reviewed your manuscript are included at the foot of this letter.

on behalf of Dr Berat Haznedaroglu (Associate Editor) and Kevin Padian (Subject Editor)
openscience@royalsociety.org

Associate Editor Comments to Author (Dr Berat Haznedaroglu):

Associate Editor: 1

Comments to the Author:

(There are no comments.)

Reviewer comments to Author:

Appendix A

Dear Handling Editor,

We are grateful for the valuable reviewer comments, which we think improved the manuscript considerably. Accordingly, we included a new data set demonstrating, that resident phages do not produce more phages, which might be costly for their host, at low salinities. This confirms our hypothesis, that indeed infecting phages are causing the strong reduction in bacterial growth at low salinities.

Please find our point by point answers below.

Thank you for considering our manuscript for publication.

Yours sincerely,
Carolin Wendling

Reviewer comments to Author:

Reviewer: 1

Comments to the Author(s)

In this paper, the authors test phage host range under varying salinities, using conditions that represent possible variation during regular fluctuations as well as under climate change. Overall, this is a well-written paper with interesting results. I have a few minor questions/concerns listed below:

Minor concerns:

1. Depending on the journal space requirements, I would recommend moving more (or all) of the supplementary methods into the main text. Similarly, I would recommend restating more of the essential method details that are described in other papers (e.g. that phages were originally induced with Mitomycin C and diluted in TM buffer). This would be more convenient for the reader.

Response: We now moved all the sections describing the methods from the ESM to the main document and include an additional section on phage extraction. We cite Wendling, Goehlich, Roth (2018) for details about phage extraction. Neither in that nor in the present study did we use mitomycin C. We simply purified the supernatant from exponentially growing bacteria, which we already mentioned in the manuscript. As suggested, we now mention that phages were diluted in TM-buffer.

2. How reliable is the growth rate reduction assay at all salinities? Is it possible that phages are more harmful at low salinity but susceptibility is actually not higher? I.e. could you have cryptic infections that do not infect at high rates or do not infect with high virulence unless at low salinity? One possibility is that the stress from the low salinity leads to a higher rate of spontaneous phage induction, and that phages do infect at 15 PSU but do so without causing significant loss of host growth. If possible, I would suggest doing PCR assays on a random subsample of both negative and positive results to confirm the growth curve assay inferences.

Response: As these are filamentous phages, phage induction as for instance described for head-tail prophages does not happen. We can thus exclude that induction *sensu stricto* differs between salinities, simply because filamentous phages can't be induced. However, it is possible that phage production differs between salinities. We thus now measured the production of resident filamentous phages for each strain and found no difference among salinities. We now say so in the manuscript, lines 164-195, lines 228-231, and lines 275 – 280.

With the current assay we cannot exclude cryptic infections, i.e. possible infections at 15 PSU which however, do not reduce bacterial growth. We point out this limitation in the revised version of the manuscript, see lines 239-242.

3. Because the 7 PSU treatments grew slower, the authors varied the timing of the infection assay to try to do infections at the same relative part of the bacterial growth curves. Looking at the supplementary figure with bacterial growth curves, low salinity not only had slower growth but prolonged growth (and in some graphs what appears to be diauxic growth). Is it possible that these changes in timing, though, might explain some of the differences in bacterial susceptibility? That is to say, might low salinity increase the time window for phage infection?

Response: We now included these thoughts in the results and discussion section and mention a possible future study to identify whether prolonged growth might increase the time window for phage infections, see lines 270-275.

4. Very minor semantics issue: In figure S6, a label refers to “number of phages causing lytic infections.” The use of “lytic” here seems inappropriate since the assay does not distinguish between reduced growth and increased mortality (and since the phages are filamentous).

Response: We agree with the reviewer and removed “lytic” from the header

Reviewer: 2

Comments to the Author(s)

I enjoyed reading this paper. An understanding of how abiotic factors influence bacteria-phage networks is one of fundamental and applied interest. I have a couple of minor suggestions:

1. I am glad that the authors have pointed out that studying the interactions over a longer (evolutionary time-scales) is required. One could predict that bacteria will adapt to counter the negative effects on fitness at lower salinities. With this point in mind I think the authors should scale back or tone down the language relating to the potential consequences of this reduced salinity in the future with respect to the evolution and emergence of pathogens i.e. last sentence of the abstract and Line 298-299 e.g. “we can expect the emergence of new pathogenic Vibrios in response to decreasing salinities.” The use of the word “can” expect sounds as though this outcome is almost guaranteed... Similarly, I think the final sentence of the abstract is overstated based on the actual findings presented.

Response: We now included a sentence saying that bacterial adaptation to increased phage pressure could be possible and tuned down the final sentence in the discussion and abstract.

2. Spell out PSU in the first instance

Response: We did spell out PSU as practical salinity units at the first instance for every section (Abstract, Introduction, Material & Methods, Results & Discussion)

3. Line 181 check font

Response: done